# Point-of-choice kilocalorie labelling practices in large, out-of-home food businesses: a preobservational versus post observational study of labelling practices following implementation of The Calorie Labelling (Out of Home Sector) (England) Regulations 2021

Megan Polden [1,2] Andrew Jones,[3] Michael Essman [4] Jean Adams,[4] Tom Bishop,[4] Thomas Burgoine,[4] Aisling Donohue,[3] Stephen Sharp,[4] Martin White,[4] Richard Smith,[5] Eric Robinson[6]

For numbered affiliations see end of article.

**Correspondence to**
Megan Polden;
m.polden@liverpool.ac.uk

## ABSTRACT

**Background and objectives** On 6 April 2022, the UK government implemented mandatory kilocalorie (kcal) labelling regulations for food and drink products sold in the out-of-home food sector (OHFS) in England. Previous assessments of kcal labelling practices in the UK OHFS found a low prevalence of voluntary implementation and poor compliance with labelling recommendations. This study aimed to examine changes in labelling practices preimplementation versus post implementation of mandatory labelling regulations in 2022.

**Methods** In August–December 2021 (preimplementation) and August–November 2022 (post implementation), large OHFS businesses (250 or more employees) subject to labelling regulations were visited. At two time points, a researcher visited the same 117 food outlets (belonging to 90 unique businesses) across four local authorities in England. Outlets were rated for compliance with government regulations for whether kcal labelling was provided at any or all point of choice, provided for all eligible food and drink items, provided per portion for sharing items, if labelling was clear and legible and if kcal reference information was displayed.

**Results** There was a significant increase (21% preimplementation vs 80% post implementation, OR=40.98 (95% CI 8.08 to 207.74), p<0.001) in the proportion of outlets providing any kcal labelling at point-of-choice post implementation. Only 15% of outlets met all labelling compliance criteria post implementation, with a minority of outlets not presenting labelling in a clear (33%) or legible (29%) way.

**Conclusion** The number of large businesses in the OHFS providing kcal labelling increased following the implementation of mandatory labelling regulations. However, around one-fifth of eligible outlets sampled were not providing kcal labelling 4–8 months after the regulations came into force, and the majority of businesses only partially complied with government guidance. More effective enforcement may be required to further improve kcal labelling practices in the OHFS in England.

**Preregistration** Study protocol and analysis strategy preregistered on Open Science Framework (https://osf.io/pfnm6/).

---

## STRENGTHS AND LIMITATIONS OF THIS STUDY

⇒ This is the first study in England to investigate kcal labelling compliance in the out-of-home food sector (OHFS) preimplementation versus post implementation of mandatory kcal labelling regulations.

⇒ This study had high inter-rater reliability when examining adherence to labelling guidance and assessed a range of OHFS businesses from multiple local authorities sampled from across the socioeconomic spectrum.

⇒ The study only examined large OHFS businesses and therefore findings are not reflective of the whole OHFS.

⇒ The current study did not examine consistency across different outlets within a chain and therefore conclusions cannot be drawn on the level of consistency of kcal labelling within large chains.

## INTRODUCTION

The prevalence of obesity has been steadily rising both globally and in the UK since 1980,[1 2] with 64% of adults and 38% of 10–11 year olds in England classified as having overweight or obesity.[3] Between 25% and 39% of UK adults reported eating food from the out-of-home food sector (OHFS) at least once per week in 2008–2012 and 2018.[4–6]

Food purchased from the OHFS is often high in energy[7 8] and frequent consumption of this food is associated with poorer diet quality and overweight/obesity.[4 9 10] A 2021 study in England found that on average people purchased 1013 kcals per meal when eating in large out of the home food outlets and consumed an average of 915 kcals,[11] significantly higher than the UK government recommendation of 600 kcals or less for a lunch or evening meal.[12] Consumers also tend to underestimate the energy content of food purchased from the OHFS.[13 14] In addition, poor-quality diet and obesity are associated with a lower socioeconomic position (SEP)[15] and people of lower SEP are more likely to report being less motivated by weight management or healthiness of food choices,[16] which may contribute to SEP-related differences in OHFS behaviour.

Multiple countries and regions, including the USA[17] and Canada,[18] have introduced mandatory policies that require OHFS outlets to provide consumers with point-of-choice kcal labelling for food and drink products. Meta-analytic evidence from randomised control trials and longitudinal studies indicates that kcal labelling may have small public health benefits, as the provision of kcal labelling has been shown to reduce the number of kcals purchased in food outlets.[19] Consistent with this, a recent US study examining purchasing transaction data found that kcal labels were associated with approximately 25 fewer kcals purchased suggesting that labelling in real-world setting may influence consumer behaviour.[20] However, other reviews have concluded that kcal labelling has a negligible effect on consumer behaviour.[21] Effects are likely mediated to some extent by the quality of labelling implemented and labels being noticed and used. Thus, high-quality implementation will likely be important to maximise the effects of kcal labelling on consumer purchasing and consumption.

In 2011, the UK government launched the public health responsibility deal with companies making voluntary pledges to improve public health—including providing kcal labelling in the OHFS and best practice guidelines on how to provide labelling.[22] The extent to which this pledge resulted in business practice change was unclear, as evaluations were reliant on self-reported and incomplete data from businesses.[23] In a 2018 study, only 18 out of 104 (17%) unique business OHFS outlets in England provided in-store kcal labelling.[23] Furthermore, the quality of this kcal labelling was inconsistent, often lacking prominence and clarity, and not presented on all available food and drink items.[24] The overall quality of labelling presented is likely to impact whether customers notice and use kcal labelling when selecting products.[25]

Motivated by the lack of voluntary kcal labelling compliance in the OHFS, the UK government consulted on mandating kcal labelling in 2018. Following public consultation,[26] legislation was passed in July 2021, with mandatory kcal labelling in large food businesses required as of 6 April 2022.[27 28] The regulations require large (250 or more employees) businesses (cafes, fast-food outlets, sit-down restaurants, pubs) in England selling food and non-alcoholic drinks in scope of the regulations to (1) display the energy content of food in kcals; (2) reference to the size of the portion to which the kcal information relates (eg, sharing portion for two people); and (3) display the statement that 'adults need around 2000 kcal a day'. Food and non-alcoholic drinks are in scope of the regulations and classed as eligible if they are (1) offered for sale in a form which is suitable for immediate consumption; (2) not prepacked food; and (3) not exempt food such as food on a temporary menu for less than 30 days.[29] The regulations state that labelling should be easily visible, clearly legible and not in any way hidden or obscured. The labelling must be displayed on the menu next to the description or price of the food. If food is on display in a food counter, the labelling must be next to or in close proximity to each food item and displayed in a position which ensures that the label can be read by the customer.[29]

To date, only one study has examined OHFS adherence to mandatory kcal labelling legislation.[30] This US study found an overall high level of compliance (94%) across 197 chain restaurants subject to federal menu labelling law (20 or more US sites). Similar data are not currently available following the implementation of regulations in England and evidence on how OHFS kcal labelling practices changed following the implementation of mandatory kcal labelling will be able to inform any future development of the regulations.

This study aimed to compare kcal labelling practices in large OHFS businesses in England subject to kcal labelling regulations before (2021) and after (2022) mandatory labelling regulations in 2022. We examined changes in the proportion of outlets providing kcal labelling and the level of compliance with government guidelines regarding the clarity, prominence and legibility of the labelling; overall and by type of outlet and local authority (LA) deprivation.

## METHODS
### Area and outlet sampling
Sampling of outlets was conducted in four purposively selected local government areas (ie, LAs) that ensured geographical coverage and representation across quintiles of deprivation (Index of Multiple Deprivation (IMD)[31] used at the LA level). LA average IMD levels (1–5) were used with IMD1 reflecting the most deprived areas and IMD5 reflecting the least deprived, defined at the lower layer super output area (LSOA) to better capture small area geographical variations in IMD. The four LAs sampled were Liverpool (IMD1, northern region), Dudley (IMD2, midlands), Milton Keynes (IMD3, South) and Richmond upon Thames (IMD5, London).

The Inter-Department Business Register[32] was used to identify eligible businesses likely to be subject to the mandatory kcal labelling policy within LAs of interest (data sampled in June 2021, list produced in Autumn

2020). This is a list of all UK businesses, which includes their core characteristics, number of employees and principal activities defined using the Standard Industrial Classification. We identified Standard Industrial Classification codes likely to include businesses serving food (for the full list of Standard Industrial Classification codes we used see online supplemental material section 1) and then identified large businesses with 250 or more employees. Individual businesses often have multiple outlets (eg, chain restaurants), so individual outlets belonging to each identified large business[33] within the four LAs of interest were identified using Ordnance Survey Points of Interest[34] data from September 2020.

The sample size was determined based on 2018 data showing that 17% of large OHFOs in the UK provided voluntary kcal labelling.[24] Based on a power calculation (see online supplemental materials section 4 for full details), to detect a doubling of prevalence from 17% to 34%, we estimated we required a minimum sample size of 96 outlets. We used stratified sampling by LA and business type to sample a total of 117 outlets, representing 90 unique businesses, due to the limited availability of unique outlets in some LAs. We prioritised sampling from unique business outlets (as opposed to sampling multiple outlets from the same business) as we assumed kcal practises would be more likely to differ between, as opposed to within, the same large businesses in the OHFS.

Three outlets closed in the post policy period and were therefore excluded from the final analysis, resulting in a sample size of n=114 (87 unique businesses). This sample included outlets from attractions, cafes, fast food, hotels, pubs, restaurants, retail and entertainment businesses (full list of businesses in online supplemental material section 2) across the four LAs.

## Procedure

The kcal labelling assessment procedure was based on previous methods used to assess kcal labelling practises of large OHFS businesses in England during 2018.[24] Researchers visited selected outlets in August–November 2021 (prepolicy) and August–November 2022 (post policy) to examine if kcal labelling was present and whether it adhered to labelling guidelines provided by the Department of Health and Social Care.[29 35 36] In each outlet, researchers examined point-of-choice menu display boards inside and outside the outlet, handheld food and drink menus and menus presented at ordering points. A researcher rated the presence of any kcal labelling (0=no, 1=yes), as well whether kcal labelling adhered to the following criteria: (1) provided at any point of choice, (2) provided at all points of choice, (3) provided for all eligible food items, (4) provided per portion for sharing items, (5) presented close to the item's name and price, (6) presented as prominently as name or price, (7) provided alongside any kcal reference information, (8) provided alongside clearly and prominently displayed kcal reference information and (9) provided for all non-alcoholic drink items (see table 1). Eligible food and

drink items include non-prepacked food that is suitable for immediate consumption. Exemptions include items on the menu for less than 30 days and drinks containing more than 1.2% by volume of alcohol.[28]

A score between 0 and 9 was calculated for each outlet based on the number of kcal labelling practices the outlet adhered to. For example, a score of 3 would mean that the outlet adhered to 3 of the labelling guidelines (list of criteria in online supplemental materials section 3). For the post implementation assessment, three additional kcal labelling criteria relating to whether labelling was clearly presented, legible and whether a kcal-free menu was available on request were added for further descriptive information (table 1) based on kcal labelling guidance introduced alongside the new legislation.[29]

Researchers received training on assessing outlet adherence to labelling guidelines and the study protocol before the start of data collection to ensure consistency between raters. Ten per cent (n=17) of outlets were randomly selected across all outlet locations and were independently coded by a second researcher with a percentage agreement score of 96% across all 9 variables for the preassessment and 97% across all 12 variables for the post assessment.

## Patient and public involvement

This study did not include patient and public involvement.

## Data analysis

A descriptive overview is presented for kcal labelling practices preimplementation and post implementation by outlet type and IMD quintile of the postcode of the location at the LSOA[37] level. Due to the small number of outlets for some outlet types (eg, entertainment and attractions), we combined categories into three meaningful categories with adequate numbers: restaurants; entertainment, retail and attractions; cafes, fast-food, pubs and hotels.

To examine whether kcal labelling practices changed from before to after implementation and whether outlet characteristics were associated with kcal labelling regulations and compliance, regression models (logistic and Poisson) were used. Exposure variables were time (preimplementation vs post implementation), outlet type (entertainment and retail venues; cafes, fast-food and pubs; and restaurants), deprivation level (LSOA IMD 1–5 (categorical)) and LA (Liverpool, Dudley, Milton Keynes, Richmond Upon Thames). Where the outcome variable was kcal labelling implemented in any form (yes/no), a logistic regression model was used. Where the outcome variable was the total compliance score (0–9), a Poisson regression model was used. Unadjusted models included only the time variable. Adjusted models additionally included outlet type, deprivation level (IMD of LSOA) and LA. We repeated these analyses limiting them to unique outlets (see online supplemental materials section 5, tables 1 and 2). In instances where data were

**Table 1** Level of compliance with kcal labelling guidelines in outlets preimplementation and post implementation.

| | Preimplementation and postimplementation assessments in all outlets* (n=117) | | Preimplementation and postimplementation assessments limited to outlets with labelling present pre and post (n=24) | |
|---|---|---|---|---|
| | Preimplementation assessment N (%) | Postimplementation assessment N (%) | Preimplementation assessment N (%) | Postimplementation assessment N (%) |
| kcal labelling provided at any point of choice | 24 (21) | 91 (80) | 24 (100) | 24 (100) |
| kcal labelling provided at all points of choice | 16 (14) | 76 (67) | 16 (67) | 22 (92) |
| kcal labelling provided for all food items | 17 (15) | 80 (70) | 17 (70) | 16 (67) |
| kcal labelling provided per portion for sharing menu items | 10 (9) | 19 (17) | 10 (42) | 7 (29) |
| kcal labelling presented close to the item's name and price | 21 (18) | 91 (80) | 21 (88) | 22 (92) |
| kcal labelling presented as prominently† as name or price | 0 (0) | 7 (6) | 0 (0) | 2 (8) |
| kcal reference information displayed anywhere | 11 (9) | 83 (73) | 11 (45) | 21 (88) |
| kcal reference information displayed clearly and prominently | 4 (3) | 51 (45) | 4 (17) | 12 (50) |
| kcal labelling provided for all non-alcoholic drink items | 11 (9) | 79 (69) | 11 (45) | 21 (88) |
| **Additional data only available for postimplementation assessment** | | | | |
| kcal labelling legible‡ | | 76 (67) | | 19 (79) |
| kcal labelling presented clearly§ | | 81 (71) | | 19 (79) |
| Non-kcal menu available on request | | 11 (12) | | 0 (0) |

*Preimplementation total outlets=117 individual outlets from across 90 unique businesses. Postimplementation total outlets=114 individual outlets from across 87 unique businesses post implementation.
†The prominence of the labelling was determined based on whether the labelling was of equal prominence as item name or price, that is, text size and colour.
‡The legibility was determined by whether the kcal labelling was clear enough to read, that is, font size, font type and colour of text.
§The clarity of the labelling was determined based on the kcal labelling was easy to understandable, that is, multiple kcal options presented close together or multiple kcal options requiring the customer to do additional sums to determine the kcal content.

available from multiple outlets within a business, compliance data were used from the first outlet in which data were collected.

In subsequent preregistered exploratory models, we planned to examine whether changes in compliance following labelling regulations differed by outlet type, LA and deprivation level. However, some of these three-way interaction analyses were not possible due to small sample sizes (Ns <5, eg, only two restaurants had kcal information at baseline and only four entertainment/retail/attraction outlets did not report kcal information at post) meaning these interaction analyses would have been severely underpowered.[38] However, we conducted simple interaction analyses between time and IMD (at both IMD of

LSOA and neighbourhood of outlet level), and time and outlet type for total compliance scores.

Finally, for the subset of outlets that had kcal labelling present at the preassessment, we performed the analysis described above (excluding interactions) to examine whether labelling regulations were associated with improvements in labelling quality among outlets already providing kcal labelling. This approach supplements the main analyses, by addressing improvements in labelling quality among existing compliant outlets, as changes in overall labelling quality in the primary analyses may be driven by outlets introducing labelling (as opposed to also increasing compliance with guidelines for how labelling is provided).

To account for the relatively large number of analyses conducted, CIs and p values were set to 99% and <0.01, respectively, to determine statistical significance. R and R Studio V.1.2.503 was used to conduct analyses, using the 'lme4', 'lmertest', 'sjPlot', 'descr' and 'performance' packages. Code and data are available here (https://osf.io/gp3rf/). The study did not involve human participants and made use of publicly available information; therefore, ethical approval was not required. The study protocol and analysis strategy were preregistered on Open Science Framework (https://osf.io/pfnm6/).

## RESULTS

Table 1 shows adherence to the policy preimplementation and post implementation of the regulations. Of 114, 91 outlets (80%) provided kcal labelling at any point of choice post implementation, an increase compared with preimplementation with 24/117 (21%). Post implementation, outlets were most compliant with presenting kcal information close to the item's name and price (80%) and displaying kcal referencing information (73%). Outlets were least compliant with presenting labelling that is as prominent as name or price (7%) and providing per portion information for sharing menu items (19%).

### Outcome: kcal labelling presented in any form

An unadjusted model with time (preimplementation vs post implementation) demonstrated a significant effect with OR=40.98 (95% CI 8.08 to 207.74), Z=4.48, p<0.001: marginal $R^2$=0.40, indicating an increased odds of kcal labelling presented in any form post implementation of about 41 times than seen preimplementation.

In an adjusted model (table 2), the effect of time remained significant. However, the IMD quintile of outlet location, outlet type and LA were not significantly associated with any kcal labelling.

There was some evidence of multicollinearity between LA (Variance Inflation Factor=3.57) and IMD quintile of LSOA (Variance Inflation Factor=4.50). Removal of either variable did not affect the models. We repeated the analyses limiting them to unique outlets. The overall pattern of results was similar; post implementation led to OR=25.64 (95% CI 3.72 to 176.87) increased odds for any kcal labelling, in the full model (see online supplemental materials section 5).

### Overall compliance score

A multilevel Poisson model with a random intercept of outlet examined overall compliance score (0–9). An unadjusted model with time (preimplementation vs post implementation) demonstrated a significant effect with IRR=5.06 (95% CI 3.90 to 6.56), Z=16.10, p<0.001: marginal $R^2$=0.49, indicating that compliance score was six times greater post implementation versus preimplementation of the regulations. Preimplementation, the mean compliance score was 1.00/9 (2.18) and post implementation it was=5.06/9 (2.80). In an adjusted model

**Table 2** Effects of labelling regulations, index of multiple deprivation, local authority and outlet type on presentation of any kcal labelling

| Exposure variables | Any kcal labelling | | |
| --- | --- | --- | --- |
| | ORs | 99% CI | P value |
| Time (post implementation) | 35.11 | 4.87 to 252.98 | **<0.001** |
| IMD-2 | 0.24 | 0.03 to 1.76 | 0.065 |
| IMD-3 | 1.17 | 0.23 to 5.86 | 0.804 |
| IMD-4 | 0.17 | 0.02 to 1.88 | 0.058 |
| IMD-5 | 0.21 | 0.02 to 1.89 | 0.067 |
| LA Liverpool | 1.43 | 0.27 to 7.65 | 0.580 |
| LA Milton Keynes | 0.76 | 0.15 to 3.79 | 0.658 |
| LA Richmond | 1.63 | 0.19 to 13.67 | 0.555 |
| Outlet type (entertainment, retail, attractions) | 0.50 | 0.09 to 2.89 | 0.307 |
| Outlet type (restaurants) | 0.49 | 0.15 to 1.65 | 0.130 |
| **Random effects** | | | |
| $\sigma^2$ | 3.29 | | |
| $\tau_{00\ outletid}$ | 1.07 | | |
| Intraclass Correlation Cefficients | 0.24 | | |
| $N_{outletid}$ | 114 | | |
| Marginal $R^2$/conditional $R^2$ | 0.465/0.596 | | |

Reference categories were preimplementation, IMD = 1; Local authority of Dudley; cafes, fast-food, pubs and hotels for outlet type.
Significant at the p<0.01 level
IMD, Index of Multiple Deprivation; LA, local authority.

(table 3), time remained significant and the IMD quintile at the LSOA level, outlet type and LA were not significant predictors of the overall compliance score.

There was some evidence of multicollinearity between LA (Variance Inflation Factor=3.50) and IMD quintile (Variance Inflation Factor=3.50). Removal of either variable from the model did not substantially influence the results. We repeated the analyses limiting them to only unique outlets. The overall pattern of results was similar; post implementation led to IRR=5.19 (95% CI 3.83 to 7.04) increase in any kcal labelling, in the full model (see online supplemental materials section 5).

In separate models, we examined the interaction between time and IMD, time and LA, and time and outlet type. There were no significant interactions between time and LA (ps >0.135). There was a significant interaction between time and outlet type, specifically restaurants (IRR=10.66 (95% CI 3.58 to 31.78), p<0.001). Compared with cafès, fast-food, pubs and hotels, there were greater increases in compliance for kcal labelling in restaurants post implementation (increase in compliance scores in

**Table 3** Effects of labelling regulations, index of multiple deprivation, local authority and outlet type on total compliance score for kcal labelling

| Total compliance score (0–9) | | | |
|---|---|---|---|
| *Exposure variables* | *Incidence rate ratios* | **99% CI** | **P value** |
| Time (post implementation) | 5.06 | 3.90 to 6.56 | **<0.001** |
| IMD-2 | 0.70 | 0.38 to 1.28 | 0.128 |
| IMD-3 | 1.10 | 0.67 to 1.80 | 0.623 |
| IMD-4 | 0.61 | 0.29 to 1.29 | 0.088 |
| IMD-5 | 0.66 | 0.34 to 1.29 | 0.110 |
| LA Liverpool | 0.99 | 0.59 to 1.65 | 0.948 |
| LA Milton Keynes | 0.85 | 0.51 to 1.40 | 0.395 |
| LA Richmond | 1.10 | 0.56 to 2.17 | 0.704 |
| Outlet type (entertainment, retail, attraction) | 0.82 | 0.48 to 1.43 | 0.366 |
| Outlet type (restaurants) | 0.94 | 0.65 to 1.36 | 0.650 |
| **Random effects** | | | |
| $\sigma^2$ | 0.33 | | |
| $\tau_{00\ outletid}$ | 0.30 | | |
| Intraclass Correlation Cefficients | 0.48 | | |
| N $_{outletid}$ Marginal $R^2$/conditional $R^2$ | 114 0.528/0.754 | | |

Reference categories were preimplementation, IMD = 1; Local authority of Dudley; cafes, fast-food, pubs and hotels for outlet type.
IMD, Index of Multiple Deprivation; LA, local authority.

restaurants=5.33, increase in cafes, fast-food, pubs and hotels=3.54). Finally, there was a significant interaction between time and IMD, specifically for IMD2 (IRR=18.73 (95% CI 1.42 to 246.54), p=0.003) and IMD5 (IRR=2.97 (95% CI 1.33 to 6.62), p<0.001). There was a larger increase in compliance scores for IMD2 (increase=4.86) and IMD5 (increase=4.34), compared with IMD1 (increase=3.92). Full model reporting can be found in online supplemental section 5, tables 3-5.

### Exploratory analysis: examining changes in compliance score among 24 outlets that displayed kcal at both time periods

We limited our analysis to the 24 outlets which had any kcal labelling at preimplementation and examined whether labelling regulations increased the total compliance score. There was no evidence that the total compliance scores significantly increased from preimplementation to post implementation in these outlets (IRR=1.25 (95% CI 0.97 to 1.59), z=1.75, p=0.081), indicating that the change in compliance score observed in the primary analysis was largely driven by more outlets providing any form of kcal labelling, as opposed to compliance to recommendations significantly increasing in outlets already providing kcal labelling. At the

preimplementation time point, mean compliance score=4.75/9 (2.17) and at post implementation, mean compliance score=5.92/9 (2.28). Frequency data on

individual labelling criteria pre versus post for the subset of 24 outlets are presented in table 1.

### Discussion

Overall, results indicated that the introduction of labelling regulations was associated with a significant increase in the likelihood of OHFS outlets providing kcal labelling, with 21% in 2021 compared with 80% in 2022 providing labelling in any form. Compliance with government guidelines for the nine kcal labelling criteria also increased post implementation. The changes indicate the policy had a significant impact on labelling practices and prevalence in the OHFS. However, it should be noted that although compliance overall increased, there was still a substantial number of eligible outlets not providing any form of kcal labelling post implementation of the policy (20%). In addition, there remained a lack of compliance with labelling guidelines, particularly in relation to presenting labelling and kcal reference information clearly and prominently; only 17/114 outlets (15%) met all guideline criteria post implementation. When analyses were limited to outlets already implementing kcal labelling preregulations, there was no evidence of a significant change in compliance scores for labelling guidelines. This finding suggests that the increase in overall compliance scores from preimplementation to post implementation observed in the full sample was strongly driven by outlets

introducing new labelling, as opposed to substantial improvements in the overall quality of existing labelling.

Twenty per cent of eligible outlets sampled were not providing any in-store kcal labelling at the time of post implementation assessment. This reflects a rate of non-compliance greater than that observed in a US study, where only 6% of eligible outlets were not implementing kcal labelling in a similar number of unique businesses.[30] Of the outlets that provided in-store kcal labelling in our study, most were not presented as prominently as name or price, presented at all points of choice, presented on all eligible food and drink items and for some outlets and not presented clearly or legibly. These results are consistent with research conducted prior to the announcement of the 2022 kcal labelling policy which found that existing voluntary kcal labelling did not align with government recommendations in the UK OHFS.[24] We also examined the legibility and clarity of kcal labelling post regulations as this was specified in the regulations. The majority of outlets sampled post regulations presented kcal labelling that was legible (67%) and clear (71%), but it often lacked prominence (94% of kcal labelling rated as not as prominent as name or price). This lack of compliance and potential hesitancy by food outlets to provide labelling information that is clear, legible and prominent may reflect an attempt by businesses to minimise the potential for kcal information to negatively impact product sales. Additionally, a lack of public support for kcal labelling policy[39] may have impacted on businesses willingness to fully comply with labelling guidance and in particular provide labelling that is prominent to customers. Food industries, specifically pizza chains in the USA, have argued for flexibility when providing labelling due to difficulties reporting kcal amounts for customisable food items with multiple combinations and frequently changing menus,[30] and this may be a contributory factor to a lack of compliance in England, particularly when reporting sharing options on menus. The level of impact of mandatory kcal labelling on consumer behaviour may be dependent on consumers' level of engagement[39] and this may be mediated by the prevalence and prominence of kcal labelling in the OHFS. Whether kcal labelling is implemented and labelling is clearly and prominently presented is likely to impact whether customers notice and use kcal labelling when selecting products.[25]

There was a significant interaction between pre–post implementation and outlet type, specifically with restaurants showing a greater increase in compliance compared with cafes, fast-food, pubs and hotels. This interaction may have been driven by the particularly low compliance levels of restaurants at the preimplementation time point compared with other outlet types. Furthermore, there was a significant interaction between time and IMD, specifically that outlets from IMD quintiles 2 and 5 showed a statistically significant larger increase in compliance scores compared with IMD1. This may indicate that outlets located in more affluent areas showed greater improvement in kcal labelling practice post regulations

although it is not clear why this association was not found for IMD quintiles 3 and 4. The sample sizes in these analyses are small and further research examining the robustness of these associations is needed to determine their validity. This difference may also have been driven by the prevalence of different outlet types in each area, for example, proportionally more fast-food establishments (as opposed to pubs) in Liverpool compared with Richmond Upon Thames.

Providing kcal information at the point of choice in the OHFS is hypothesised to support consumers in making informed dietary decisions and some, but not all, evidence indicates that this leads to consumers purchasing lower kcal and healthier food options.[40] As a proportion of major OHFS businesses are not complying with the kcal labelling regulations, this may limit the impact on consumer behaviour.

Recent qualitative data suggested that LAs are not engaging in proactive enforcement, and instead relying on the presumed compliance of large businesses who they expect to comply with regulations to protect their reputations and maintain customer trust.[41] However, our study has found that 20% of eligible outlets are not adhering to kcal labelling regulations in any form and 85% are not complying with labelling guidance in full. Our findings suggest that greater enforcement by LAs may be required to improve current labelling practices in the UK OHFS. It may also be the case that more specific details and regulations are needed in relation to how labelling should be presented within outlets, for example, minimum font size, colour and font type. This may aid in improving the legibility, clarity and prominence of labelling presented and also lead to greater consistency across businesses.

There has been some public concern about the implementation of mandatory kcal labelling in England,[39] specifically about the potential negative impacts on people with eating disorders.[42] In an attempt to mitigate potential negative impacts, as part of government guidance for kcal labelling, it was recommended that kcal-free menus be available in outlets on request. Our results found that only 12% of outlets implementing kcal labelling were able to provide researchers with a kcal-free menu when requested in-store. Therefore, attempts to mitigate the potential harms of kcal labelling on vulnerable groups may require more formal guidance or a mandate to make the practice of kcal-free menus available on request more widespread.

### Strengths and limitations

To our knowledge, this is the first study that has assessed and compared kcal labelling practices in the OHFS before and after the implementation of mandatory kcal labelling regulations in England. Data collection for kcal labelling assessments had high inter-rater reliability between researchers. This study assessed a range of OHFS businesses from multiple LAs sampled from across the socio-economic spectrum in England; however, it is not clear how representative this sample is. As this study focused

on large businesses subject to the kcal labelling policy, we did not examine kcal labelling practices in outlets exempt from the policy, including smaller businesses (<250 employees) and independent outlets, so this study does not give a full representation of kcal labelling practices across all businesses in the OHFS. Although encouraged, it is currently not mandatory for smaller OHFS businesses to implement kcal labelling. The present study provides data on those businesses where kcal labelling is mandated due to the legislation. It is therefore unclear whether the policy also impacted on kcal labelling practises in outlets currently exempt from the legislation. Furthermore, it should be noted that exploratory analyses examining local area and outlet type variability in pre–post implementation changes to labelling were based on small sample sizes.

Additionally, this study examined labelling practices at a single time point post implementation (approximately 4–8 months post implementation) and it could be argued that some outlets needed more time to implement the regulations and comply with guidance. However, the government consulted on mandating kcal labelling in 2018 with a public consultation occurring in 2020. The legislation was then passed in 2021 with labelling being required from April 2022. This timeline of events provided businesses with ample time to prepare and enact required regulations. Therefore, it is unlikely that businesses did not have sufficient time to implement labelling regulations ahead of this labelling assessment and it is unlikely that this is a legitimate reason for a lack of or poor compliance in some outlets in this assessment.

## Future research

Our study examined large businesses in the OHFS subject to mandatory kcal labelling regulations. There has been debate as to whether to expand these regulations to include medium and small OHFS businesses and future research could examine whether kcal labelling regulations also motivated smaller businesses not subject to the policy to provide voluntary kcal labelling on menus. Additionally, we were unable to examine whether labelling is provided consistently within outlets of the same business and this could be explored in future research. Research is also yet to examine the nutritional accuracy of newly implemented kcal labelling in the OHFS in England and future research would benefit from examining this.

## Conclusions

Overall, the findings demonstrate a significant increase in kcal labelling practices in the OHFS in England following the implementation of the labelling regulations. However, there was still a substantial proportion of outlets not complying with the kcal labelling post implementation, not fully meeting labelling guidance and providing kcal labelling in a way that undermines the likelihood of it being used by consumers. More proactive enforcement may be required to further improve kcal labelling practice in the UK OHFS and businesses should be encouraged to review their labelling practices to ensure continued compliance with labelling regulation guidelines.

**Author affiliations**
[1]Department of Primary Care and Mental Health, University of Liverpool, Liverpool, UK
[2]Lancaster University, Lancaster, UK
[3]Liverpool John Moores University, Liverpool, UK
[4]MRC Epidemiology Unit, University of Cambridge, Cambridge, UK
[5]Institute of Health Research, University of Exeter Medical School, Exeter, UK
[6]Department of Psychology, University of Liverpool, Liverpool, UK

**Contributors** Funding was obtained by JA, ER, AJ, MW, RS, TB and SS. Conceptualisation and methodology design contributions and project administration contributions were made by all authors. The formal analysis, data curation and original draft preparation contributions were made by MP, ER and AJ. All authors contributed to revising the manuscript. All authors read and agreed to the published version of the manuscript. The guarantors for this manuscript are MP and ER.

**Funding** This report is independent research commissioned and funded by the Department of Health and Social Care Policy Research Programme (Implementation and assessment of mandatory kcal labelling in the out-of-home sector, NIHR200689). The views expressed in this publication are those of the author(s) and not necessarily those of the NIHR or the Department of Health and Social Care. JA, TB, TB, ME, SS and MW are supported by the MRC Epidemiology Unit, University of Cambridge (UKRI grant number MC/UU/00006/7). MP receives support from the NIHR Applied Research Collaboration ARC NWC and Alzheimer's Society and is funded through a Post-Doctoral Fellowship. ER is funded by the National Institute for Health and Care Research (NIHR) Oxford Health Biomedical Research Centre (BRC), Economic and Social Research Council (ES/W007932/1) and European Research Council (Grant reference: PIDS, 803194). The views expressed are those of the authors and not necessarily those of the funders, NHS or Department of Health and Social Care. For the purpose of Open Access, the author has applied a Creative Commons Attribution (CC BY) licence to any Author Accepted Manuscript version arising.

**Competing interests** ER has previously received funding from Unilever and the American Beverage Association for unrelated research. AJ has previously received funding from Camurus pharmaceuticals, unrelated to this project

**Patient and public involvement** Patients and/or the public were not involved in the design, or conduct, or reporting, or dissemination plans of this research.

**Patient consent for publication** Not applicable.

**Ethics approval** Not applicable.

**Provenance and peer review** Not commissioned; externally peer reviewed.

**Data availability statement** All data relevant to the study are included in the article or uploaded as supplementary information.

**ORCID iDs**
Megan Polden http://orcid.org/0000-0002-1813-0765
Michael Essman http://orcid.org/0000-0001-5017-3880

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
