## [Reviewer comments · BMJ Open]

ARTICLE DETAILS

TITLE (PROVISIONAL)	Point-of-choice kilocalorie labelling practices in large, out-of-home food businesses: A pre vs post observational study of labelling practices following implementation of kilocalorie Labelling (Out of Home Sector) (England) Regulations 2021
AUTHORS	Polden, Megan; Jones, Andrew; Essman, Michael; Adams, Jean; Bishop, Tom; Burgoine, Thomas; Donohue, Aisling; Sharp, Stephen; White, Martin; Smith, Richard; Robinson, Eric

VERSION 1 – REVIEW

REVIEWER	Prates, Sarah Morais Senna Universidade Federal de Minas Gerais, Food Science
REVIEW RETURNED	05-Dec-2023

GENERAL COMMENTS	I thank you for the opportunity to review this paper and congratulate the authors on the quality of their work. I have a few suggestions to make it more straightforward for the reader. Below are my comments and suggestions: INTRODUCTION: Page 4, line 8: “38% of 10–11-year-olds” Why present data only for this age group (10-11 years old) and not for a broader age group? Page 4, lines 11-12: “Between 25% and 39% of UK adults reported eating food from the out-of-home food sector (OHFS) at least once per week in 2008-12 and 20184-6”. Exist more recent data? The COVID-19 pandemic may have influenced this food consumption outside the home. Page 5, line 40: “(2) reference the size of the portion to which the kcal information relates.” Is the portion size random (at the company's discretion), or is there a standardization to the portion size depending, for example, on the product type?
--

Page 5, second paragraph:

In this paragraph, it is worth specifying the guidelines regarding clarity, prominence, and readability of kcal labeling since compliance with these criteria was assessed in the study.

For example, what is considered clear labeling?

Page 6, lines 15-17: "government guidelines regarding the clarity, prominence and legibility of the labelling"

As in the previous comment, it is worth specifying more clearly these guidelines (in the 4th paragraph of the introduction). What would be highlighted information? Bold, capital letters?

METHODS

Page 6, lines 38-42:

What about IMD 4?

RESULTS

I suggest that you make a brief presentation of the data in Table 1. For example, which criteria are the most compliant, and which are the most neglected by establishments?

Tables 2 and 3: The legend must include the time ref category (pre-implementation).

DISCUSSION

This subtitle ("Summary of main findings") is unnecessary.

Page 16, lines 17-21:

I think you should remove this first sentence; I found it repetitive. I think keeping this information only in "strengths" is more interesting.

	Page 16, lines 38-44: "...post-implementation." I believe that here it is worth discussing a little more about the main criteria that are neglected (Table 1) by establishments, for example, "Kcal labeling presented as prominently as name or price"; "Kcal labeling provided per portion for sharing menu items" and "Kcal reference information displayed clearly and prominently." Why does this happen? Discuss based on literature. Pages 17-18, lines 58-3: "This may indicate that outlets located in more affluent areas showed more significant improvement in kcal labelling practice post-regulations." But why didn't this happen in IMD 3 and 4? Link with the following sentence ("However, the sample sizes...") Future Research ...to examine the nutritional accuracy: This is very interesting and fundamental! Unfortunately, even when implemented on a mandatory basis, many companies often neglect nutritional labeling guidelines.
--	--

REVIEWER	Daley, Amanda Loughborough University, School of Sport, Exercise and Health Sciences
REVIEW RETURNED	12-Jan-2024

GENERAL COMMENTS	General comments This is a well written manuscript that address a very timely question. I only have a few specific questions for the authors to consider as listed below. Since this manuscript was submitted for review a trial by Rummo et al (JAMA Open) has been published and the results of this trial may need to be considered by the authors. Specific comments Abstract Line 31: What are 'eligible food and drinks'? I didn't see reference to this in the methods? Methods Page 8 Line 56: Reference required for the Department of Health and Social Care so that readers can find this information. Page 9 Line 40: How were the 10% outlets selected for the rater validity checks? Was it a random 10% selection across all the locations? Discussion Page 19: Based on these results is appears it is not sufficient for legislation to simply state calorie labelling must in be in place and
--

	there is also need for some directions about how this is actually displayed/implemented? Strengths and limitations Page 19 line 27: The study included larger size outlets where it is much more likely that such establishments will be compliant with their lawful requirements regarding calorie labelling. Therefore, this study presents a ‘best case scenario’ – non-compliance is likely to be higher in smaller establishments?
--	--

VERSION 1 – AUTHOR RESPONSE

Reviewer 1

Introduction

Reviewer comment: Page 4, line 8: “38% of 10–11-year-olds”

Why present data only for this age group (10-11 years old) and not for a broader age group?

Author Response: The UK Office of National Statistics for the period 2021-2022 only reported for children ages between 4-5 years old (reception) and 10-11 years old (year 6) as an updated indicator of overweight and obesity levels in children following the 2019 Health Survey for England. The most recent Health Survey for England is not yet available and due to this we opted to present the data for this age group in the manuscript rather than more outdated reported obesity levels from 2019 across all age groups.

Reviewer comment: Page 4, lines 11-12: “Between 25% and 39% of UK adults reported eating food from the out-of-home food sector (OHFS) at least once per week in 2008-12 and 2018”. Exist more recent data? The COVID-19 pandemic may have influenced this food consumption outside the home.

Author Response: We are not aware of more recent data on reported frequency of eating out of the home and the reported information is currently the most up to date information available.

Reviewer comment: Page 5, line 40: “(2) reference the size of the portion to which the kcal information relates.” Is the portion size random (at the company's discretion), or is there a standardization to the portion size depending, for example, on the product type?

Author Response: This statement refers to the recommended portion size determined by the food outlet. For example, the outlet may state a food item is recommended for two people with the calories stated however there is no criteria in the regulations that specifies what this portion size should be just that it is reported to the customer. We have not altered the manuscript as this was the wording of the criteria used.

Reviewer comment: Page 5, second paragraph: In this paragraph, it is worth specifying the guidelines regarding clarity, prominence, and readability of kcal labeling since compliance with these criteria was assessed in the study. For example, what is considered clear labeling?

Author Response: Thank you for this comment. We have added in details to page 6, stating the

wording of the regulations in relation to what they state is clear and legible labelling. The following statement was added:

“The regulations state that labelling should be easily visible, clearly legible and not in any way hidden or obscured. The labelling must be displayed on the menu next to the description or price of the food. If food is on display in a food counter the labelling must be next to or in close proximity to each food item and displayed in a position which ensures that the label can be read by the customer²⁸.”

Reviewer comment: Page 6, lines 15-17: “government guidelines regarding the clarity, prominence and legibility of the labelling”. As in the previous comment, it is worth specifying more clearly these guidelines (in the 4th paragraph of the introduction). What would be highlighted information? Bold, capital letters?

Author Response: We have added more details on the wording of the government guidelines in relation to how the labelling should be displayed in the food outlet (on menus and in food display counters) (page 5-6).

“The regulations state that labelling should be easily visible, clearly legible and not in any way hidden or obscured. The labelling must be displayed on the menu next to the description or price of the food. If food is on displayed in a food counter the labelling must be next to or in close proximity to each food item and displayed in a position which ensures that the label can be read by the customer²⁸.”

Methods

Reviewer comment: Page 6, lines 38-42:
What about IMD 4?

Author Response: To identify local authorities we used IMD calculated at the local authorities level, so each local authorities only represents one IMD quintile. We were only able to sample 4 local authorities despite there being 5 IMD quintiles so we didn't have an local authorities with an local authorities -level IMD in quintile 4. However, IMD4 is represented across all outlet areas at a lower layer super output area level as across each area there is a range of IMD levels.

We have edited this section below to include this detail:

“Local authority average IMD levels (1-5) were used with IMD1 reflecting the most deprived areas and IMD5 reflecting the least deprived, defined at the lower layer super output area to better capture small area geographic variations in IMD. The four LAs sampled were Liverpool (IMD1 northern region), Dudley (IMD2 midlands), Milton Keynes (IMD3, South) and Richmond upon Thames (IMD5 London).”

Results

Reviewer comment: I suggest that you make a brief presentation of the data in Table 1. For example, which criteria are the most compliant, and which are the most neglected by establishments?
Tables 2 and 3: The legend must include the time ref category (pre-implementation).

Author Response: Thank you, we have added a short summary above the table highlighting the criteria that outlets most and least complied with. The paragraph below has been added to the manuscript on page 11. Thank you for highlighting this, we have added this information to table 2 and 3.

“91/114 outlets (80%) provided kcal labelling at any point of choice post-implementation, an increase

compared to pre-implementation with 24/117 (21%). Post-implementation, outlets were most compliant with presenting kcal information close to the item's name and price (80%) and displaying kcal referencing information (73%). Outlets were least compliant with presenting labelling that is as prominent as name or price (7%) and providing per portion information for sharing menu items (19%)."

Discussion

Reviewer comment: This subtitle ("Summary of main findings") is unnecessary.

Author Response: We have removed this subheading.

Reviewer comment: Page 16, lines 17-21: I think you should remove this first sentence; I found it repetitive. I think keeping this information only in "strengths" is more interesting.

Author Response: We have removed the following section from the beginning of the discussion section. "This is the first study we are aware of to examine point-of-choice kcal labelling practices in the OHFS before and after the implementation of mandatory kcal labelling regulations in England."

Reviewer comment: Page 16, lines 38-44: "...post-implementation."

I believe that here it is worth discussing a little more about the main criteria that are neglected (Table 1) by establishments, for example, "Kcal labeling presented as prominently as name or price"; "Kcal labeling provided per portion for sharing menu items" and "Kcal reference information displayed clearly and prominently." Why does this happen? Discuss based on literature.

Author Response: Thank you for this comment. We have added more information on why we think there may be a lack of compliance for specific criteria in particular displaying labelling and reference information in a prominent way. The following section has been added to the manuscript on page 18 when we are discussing in more detail particular aspects of the kcal labelling criteria that outlets did not comply with.

"This lack of compliance and potential hesitancy by food outlets to provide labelling information that is clear, legible and prominent may reflect an attempt by businesses to minimise the potential for kcal information to negatively impact product sales. Additionally, a lack of public support for kcal labelling policy (Polden et al, 2023) may have impacted on businesses willingness to fully comply with labelling guidance and in particular provide labelling that is prominent to customers. Food industries, specifically pizza chains in the US, have argued for flexibility when providing labelling due to difficulties reporting kcal amounts for customisable food items with multiple combinations and frequently changing menus²⁹, and this may be a contributory factor to a lack of compliance in England, particularly when reporting sharing options on menus."

Reviewer comment: Pages 17-18, lines 58-3: "This may indicate that outlets located in more affluent areas showed more significant improvement in kcal labelling practice post-regulations." But why didn't this happen in IMD 3 and 4? Link with the following sentence ("However, the sample sizes...")

Author Response: Thank you for this comment, it is not clear from the data why this association was not found in IMD3 and IMD4. We have added more information to make this clear to the reader and further highlight that this association requires further investigation to determine its validity. The section below has been amended in the manuscript (page 19):

"This may indicate that outlets located in more affluent areas showed greater improvement in kcal

labelling practice post-regulations. Although it is not clear why this association was not found for IMD quintiles 3 and 4. The sample sizes in these analyses are small and further research examining the robustness of these associations is needed to determine their validity. This difference may also have been driven by the prevalence of different outlet types in each area, for example proportionally more fast-food establishments (as opposed to pubs) in Liverpool compared to Richmond Upon Thames.”

Reviewer comment: Future Research ...to examine the nutritional accuracy: This is very interesting and fundamental! Unfortunately, even when implemented on a mandatory basis, many companies often neglect nutritional labeling guidelines.

Author Response: Thank you for this comment, we agree that it is important to determine whether the labelling present to customer is accurate and think that it could be the case that companies may not be providing accurate labelling information.

Reviewer 2

This is a well written manuscript that address a very timely question. I only have a few specific questions for the authors to consider as listed below.

Reviewer comment: Since this manuscript was submitted for review a trial by Rummo et al (JAMA Open) has been published and the results of this trial may need to be considered by the authors.

Author Response: Thank you for bringing our attention to this recently published study. We have added this study to the introduction. The following statement has been added (page 4):

“Consistent with this, a recent US study examining purchasing transaction data found that kcal labels were associated with approximately 25 fewer kcals purchased suggesting that labelling in real-world setting may influence consumer behaviour (Rummo et al, 2023).”

Abstract

Reviewer comment: Line 31: What are ‘eligible food and drinks’? I didn’t see reference to this in the methods?

Author Response: We provided information on food and drinks included in the scope of the regulations in the introduction but not in the methods. We have added further information in the methods defining what is classed as eligible food items by the regulations. The following statement has been added to the methods section on page 8.

“Eligible food and drink items include non-prepacked food that are suitable for immediate consumption. Exemptions include items on the menu for less the 30 days and drinks containing more than 1.2% by volume of alcohol²⁸.”

Methods

Reviewer comment: Page 8 Line 56: Reference required for the Department of Health and Social Care so that readers can find this information.

Author Response: Thank you for noticing this, we have added a reference.

Reviewer comment: Page 9 Line 40: How were the 10% outlets selected for the rater validity checks?

Was it a random 10% selection across all the locations?

Author Response: Yes, this is correct. The outlets were selected at random across all outlet locations. The following statement has been amended in the manuscript (page 9):

“Ten per cent (n=17) of outlets were randomly selected across all outlet locations and were independently coded by a second researcher with a percentage agreement score of 96% across all nine variables for the pre-assessment and 97% across all 12 variables for the post-assessment.”

Discussion

Reviewer comment: Page 19: Based on these results it appears it is not sufficient for legislation to simply state calorie labelling must be in place and there is also need for some directions about how this is actually displayed/implemented?

Author Response: There has been some guidance provided by the Department of Health and Social Care in relation to how it should be presented. However, it may be the case that more precise details are needed in the regulations in relation to font size, colour etc to ensure the legibility, clarity and prominence of the labelling presented and to create more consistency in presentation across outlets. The following statement has been added to the manuscript (page 19-20):

“It may also be the case that more specific details and regulations are needed in relation to how labelling should be presented within outlets, for example minimum font size, colour, font type. This may aid in improve the legibility, clarity and prominence of labelling presented and also lead to greater consistency across businesses.”

Strengths and limitations

Reviewer comment: Page 19 line 27: The study included larger size outlets where it is much more likely that such establishments will be compliant with their lawful requirements regarding calorie labelling. Therefore, this study presents a ‘best case scenario’ – non-compliance is likely to be higher in smaller establishments?

Author Response: Thank you for this comment. Yes, we agree that labelling practices in smaller OHFS businesses will be poorer as currently the policy only applies to larger businesses in England and smaller businesses are only encouraged to implement calorie labelling. We have added more information to discuss and reflect this in the manuscript. The following statement was added to page 20-21.

“Although encouraged, it is currently not mandatory for smaller OHFS businesses to implement kcal labelling. The present study provides data on those businesses where kcal labelling is mandated due to the legislation. It is therefore unclear whether the policy also impacted on kcal labelling practises in outlets currently exempt from the legislation.”

Thank you again for your helpful comments aiding us to improve the manuscript.

Yours Sincerely,
Dr Megan Polden (on behalf of all authors)

VERSION 2 – REVIEW

REVIEWER	Prates, Sarah Morais Senna Universidade Federal de Minas Gerais, Food Science
REVIEW RETURNED	08-Mar-2024
GENERAL COMMENTS	Thank you, once again, for the opportunity to review this manuscript! Congratulations to the authors for their quality work!